# Palm Raceme as a Promising Biomass Precursor for Activated Carbon to Promote Lipase Activity with the Aid of Eutectic Solvents

**DOI:** 10.3390/molecules27248734

**Published:** 2022-12-09

**Authors:** Khalid M. Abed, Adeeb Hayyan, Amal A. M. Elgharbawy, Hanee F. Hizaddin, Mohd Ali Hashim, Hassimi Abu Hasan, Mahar Diana Hamid, Fathiah M. Zuki, Jehad Saleh, Ahmad GH Aldaihani

**Affiliations:** 1Department of Chemical Engineering, Faculty of Engineering, Universiti Malaya, Kuala Lumpur 50603, Malaysia; 2Department of Chemical Engineering, College of Engineering, University of Baghdad, Baghdad 10071, Iraq; 3University of Malaya Centre for Ionic Liquids (UMCiL), Universiti Malaya, Kuala Lumpur 50603, Malaysia; 4International Institute for Halal Research and Training (INHART), International Islamic University Malaysia, Kuala Lumpur 53100, Malaysia; 5Bioenvironmental Engineering Research Centre (BERC), Department of Biotechnology Engineering, Faculty of Engineering, International Islamic University Malaysia (IIUM), Kuala Lumpur 53100, Malaysia; 6Department of Chemical and Process Engineering, Faculty of Engineering and Built Environment, Universiti Kebangsaan Malaysia, Bangi 43600, Malaysia; 7Research Centre for Sustainable Process Technology (CESPRO), Faculty of Engineering and Built Environment, Universiti Kebangsaan Malaysia, Bangi 43600, Malaysia; 8Chemical Engineering Department, King Saud University, P.O. Box 800, Riyadh 11421, Saudi Arabia; 9School of Engineering, University of Liverpool, Liverpool L69 3GH, UK

**Keywords:** activated carbon, palm racemes, carbonization, lipases immobilization, hydrolysis, deep eutectic solvents, kinetics

## Abstract

This study concerns the role of activated carbon (AC) from palm raceme as a support material for the enhancement of lipase-catalyzed reactions in an aqueous solution, with deep eutectic solvent (DES) as a co-solvent. The effects of carbonization temperature, impregnation ratio, and carbonization time on lipase activity were studied. The activities of Amano lipase from *Burkholderia cepacia* (AML) and lipase from the porcine pancreas (PPL) were used to investigate the optimum conditions for AC preparation. The results showed that AC has more interaction with PPL and effectively provides greater enzymatic activity compared with AML. The optimum treatment conditions of AC samples that yield the highest enzymatic activity were 0.5 (NaOH (g)/palm raceme (g)), 150 min, and a carbonization temperature of 400 °C. DES was prepared from alanine/sodium hydroxide and used with AC for the further enhancement of enzymatic activity. Kinetic studies demonstrated that the activity of PPL was enhanced with the immobilization of AC in a DES medium.

## 1. Introduction

Palm biomass, such as empty fruit bunches, mesocarp fiber, palm kernel shells, fronds, and trunks are under-utilized in industry. They are disposed of haphazardly by open burning or land-filling [1]. It should, instead, be turned into activated carbon (AC), which is useful for various applications [2,3,4].

In industrial-scale applications, activated carbon has been used to immobilize enzymes. Rao et al. discovered the functionality of invertase hydrolyses on charcoal adsorption in the production of enzyme nanoparticles [5]. Enzyme-supporting particle complexes have been studied to determine how the shape, size, and structure of enzymes and supporting material change [6]. The key characteristics of AC are its high porosity, its adsorption capability, and its unique surface area, which make it a suitable sorbent for the removal of various compounds [7,8,9]. Suitable carbon support material for enzyme immobilization can minimize diffusion limitations required for efficient bio-catalyzed reactions [5]. However, the high cost of commercially available AC makes it unsuitable for large-scale operations [10]. As a result, low-cost, naturally occurring supporting materials are of interest. Locally available supporting materials with low cost and excellent properties of immobilization have been given much attention in recent years. Tobacco stem, coconut husk, *Albizia lebbeck*, cotton stalk, and pineapple have been thermally activated with KOH and K_2_CO_3_ activators, yielding activated carbons with a high micropore content [9,11].

Recently, solvents such as deep eutectic solvents (DESs) have emerged, which serve as promising dual support materials for the immobilization of enzyme. The scientific and industrial communities have recently been captivated by new developments in novel deep eutectic solvents (DESs). A hydrogen bond acceptor (HBA), such as phosphonium or ammonium salt, and a hydrogen bond donor (HBD), such as ethylene glycol, urea, or glycerol, can readily be used for the preparation of DESs [12,13]. A stable product mixture and a solid network of hydrogen bonds, which offer additional characteristics, show the eutectic merits of the prepared DESs [14,15]. Adaptive freezing point, viscosity, improved solvation power and freezing point, and high thermal stability are among the physicochemical properties of DESs which are useful in industrial applications [16]. DESs also possess other merits, particularly because of their low cost and easily prepared precursor materials, resulting in cheap mixtures which operate differently [15]. Furthermore, DES storage is low-maintenance and does not require any additional purification steps. DESs can be formulated for a particular application due to their adaptive physicochemical characteristics [17]. Several research studies have been conducted on DESs and their applicability in several research fields, including chemical reactions and the manufacturing of pharmaceuticals [18]. Applications also include biofuel development [17,19], carbon nanomaterials functionalization [20,21], extraction of aromatic compounds from aliphatic mixture [22,23], porous materials synthesis [20], and biomass processing [24].

Enzymes are excellent for high-performance analysis and can be employed to facilitate environmentally-friendly and cost-effective research as natural biocatalysts [25]. Enzymes are also biodegradable and safe, and microbes may produce enormous amounts of them. Despite the benefits, Liu and Dong claim that the poor stability and reusability of free enzymes greatly restricts their ability to catalyze reactions [26]. Therefore, different diffusion restrictions imposed by substrate and its product for efficient catalyzed reactions could be overcome by immobilizing enzymes onto the support [27]. In fact, lipases can only become active when exposed to support material [28].

The supporting materials and solvents such as AC and DESs serve as a significant dual support material for the immobilization of enzymes. AC and DESs have advantages and merits for balancing important influences in boosting biocatalyst performance. Considering the excellent compatibility of AC and DES towards enzymes, the objectives of this study are: (1) to examine the influence of conditions used in preparing activated carbon on the activity of immobilized lipases of porcine pancreas (PPL) and Amano lipase (AML); (2) to determine the effects of AC and DESs on the enzymatic activity of lipase; (3) to discuss and determine the kinetics parameters of the hydrolysis of *p*-nitrophenyl palmitate by immobilized lipases at optimum reaction temperature and water content in DESs.

## 2. Results

### 2.1. Effect of Activated Carbon on Lipase Activity

#### 2.1.1. Screening of Lipases Immobilization in Different Activated Carbon

Figure 1 and Figure 2 present the relative activity of AML and PPL after the immobilization on AC, after incubating all samples for 2 h before conducting the lipase assay. PPL and AML controls (CTRL) were added for comparison (CTRL is the free forms of the enzymes PPL and AML in phosphate buffer). From the one-way ANOVA analysis, the results (Table 1) showed that R-squared is 0.9980, and there was a significant difference among means (*p* < 0.05). This implies that values can be selected based on the highest value recorded. However, we observe that AML was not enhanced by the immobilization on AC, though the activity was maintained around 80%. However, the highest red shade was observed in A1 and A2.

Figure 2 shows the effect of 13 types of AC on porcine lipase. It can be seen that A1 was recorded to have the highest activity. It also shows that the enzyme was immobilized successfully by activated carbon. The pores and active sites on the activated carbon enhanced the immobilization of enzymes in the micro- and mesopores. Furthermore, the lipase molecules are bound on the surface of the supporter after immobilization [26].

To confirm this, we have conducted a one-way ANOVA analysis to evaluate the statistical differences. R-squared revealed a good fit of the data with a significant difference among the mean values.

Table 1 supports the choice of PPL as the model enzyme for this study based on the F value obtained. The F-value in an ANOVA is calculated as the variation between sample means or variation within the samples. The higher the F-value in an ANOVA, the higher the variation between sample means relative to the variation within the samples. It can be observed that the F-value for PPL is higher than AML.

#### 2.1.2. Carbonization Temperature Effect

The surface morphology, surface area, and pore size of AC were all affected by activation time and temperature. The relative activity of two types of lipases immobilized on activated carbon prepared at various temperatures is reported in Figure 3. The carbonization temperature for all samples of AC was in the range of 400–800 °C. The enzymatic activity without AC was taken as 100% (control samples). Subsequently, Figure 3 illustrated that AML showed no improvement in enzymatic activity when immobilized on AC, while porcine lipase PPL showed more interaction with AC. As shown in Figure 3, the carbonization temperature of AC at 400 °C resulted in the highest activity for the porcine lipase. It is also worth noting that, at 700 °C, the results improved slightly. However, from a saving energy point of view, a low temperature is recommended. Therefore, 400 °C was selected as the best carbonization temperature to prepare AC in the enzymatic reaction.

A high temperature will affect the surface of the AC and may enhance the porosity of AC, and this lipase enzyme will not be bound or supported for a further reaction such as hydrolysis. For instance, Tsai [27] investigated the influence of temperature on the adsorption of chlorinated volatile organic compounds, claiming that, when the temperature was raised from 283 to 313 K with a 440 mg L^−1^ intake, the absorption of methylene chloride on AC was decreased by 60% [27]. This supports our argument for lower lipase adsorption on the AC carbonized at a high temperature above 500 °C.

#### 2.1.3. Impregnation Ratio Effect

The impregnation ratio is one of the main factors which influence the formation of porosity and the development of the surface area in activated carbon. Figure 4 shows the impregnation ratio of an activator such as NaOH to biomass (palm raceme). At a carbonization temperature of 600 °C and a carbonization time of 90 min, the impregnation ratio (IR) effect on the immobilization of lipases was investigated. At different impregnation ratios, porcine lipase (immobilized PPL) activity was enhanced compared to AML.

The impact of the activation on the ratio of raw material on the Brunner Emmet and Teller (BET) surface area, and total pore volume, was more pronounced at 400 °C. At low activation temperatures, the activator to biomass ratio substantially affected the BET surface area, micropore volume, and micropore surface area [7]. Furthermore, PPL activity was recorded at its highest at a 0.5 impregnation ratio (A10). When the impregnation ratio is higher, more Na molecules diffuse into the pores, making the holes bigger and creating large pores [28], and this is not suitable for enzyme immobilizing and may affect the reaction negatively. Thus, the activation mechanism plays a crucial role in pore growth by increasing the impregnation ratio, resulting in a sustained increase in the BET surface area and pore volume [29].

#### 2.1.4. Effect of Carbonization Time

Surface morphology, surface area, and pore size are all affected by carbonization time. Figure 5 shows the impact of carbonization time of AC on the immobilized lipase’s activity. The carbonization time for all AC samples was about 30–150 min. For the enzymatic activity of lipase, at a constant activation temperature, the influence of changing the activation time was detected. Without AC, enzyme activity was 100%. Following that, Figure 5 showed that AML recorded no enhancement in the enzymatic activity (85%), while PPL interacted with AC more effectively, providing greater and more stable adsorption with AC; this is mainly due to evaporation of volatile materials from the pores and surface of AC [30]. At the same time, new pores and active sites formed on the surface of the carbon, which can contribute to lipase immobilization. The following conditions were optimal for preparing AC for PPL lipase immobilization: an impregnation ratio of 0.5 (NaOH/palm raceme, (g/g)) and activation temperature of 400 °C for 150 min.

### 2.2. Activated Carbon and DESs Effect on the Activity of Lipases

#### 2.2.1. Effect of Reaction Temperature

Further research into the impact of temperature on the immobilized enzyme activity was carried out. The findings revealed a typical enzyme activity pattern, with the ideal reaction temperatures ranging from 40 to 80 °C for each lipase. Figure 6 depicts the behaviors of lipase at different incubation temperatures. The phosphate buffer medium was the main medium for the enzyme reaction. The findings indicate a positive tendency of changing the optimal temperatures for the reactions for two hours, from 40 °C to 80 °C. The optimum temperatures for AC/DES/Enzyme and AC/Lipase enzyme were 60 and 40 °C, respectively. A significant relative activity (290%) was achieved by incubating the lipase with AC/DES.

As the reaction temperature was raised, enzyme activity gradually increased until it reached a certain point (around 60 °C). However, the reaction rate slowed significantly due to protein denaturation at higher temperatures [31]. At high temperatures, the heat-induced disruption of non-covalent bonds triggered the partial unfolding of the enzyme. Further, because most known enzymes are proteins, they are protected from dehydration by water molecule layers which are bound to the surface of the protein. This hydrating layer, or at least a fraction of it, is a necessary component of the structure of the protein which is to serve as an enzyme [32]. Raising the temperature or adding organic solvents could significantly change the protein composition, leading to denaturation. As a result of many water molecules being withdrawn from the protein’s hydrating layer, exposing the enzyme to elevated temperature, the enzyme is inactivated. This is apparent because the enzyme’s relative activity hit 290.20% even at 70 °C when DES and AC were added as the necessary water layer required for the enzymatic activity, which is preserved due to the protein’s adsorption on AC and AC/DES complex. AC-DES exhibited good stability at an elevated temperature, as the DES was acting as additional coating and protecting the enzyme from denaturing. For instance, the experimental results indicated that lipase incubation in slightly hydrated alanine [5% (*v*/*v*)] led to the highest level of residual activity, implying interfacial activation [33]. Furthermore, several researchers have reported that glycerol-containing DESs promote lipase activity [34,35].

#### 2.2.2. Impact of Water Content

The fundamental factor which distinguishes molecular behavior in different media is the effect of water hydration between the enzyme and the bulk solvent [36]. The reaction was carried out in the DES as a medium without changing the water content of the DES in the previous section (Section 3.2). However, according to previous studies, water is needed both for the three-dimensional arrangement of enzymes and the stabilization of protein [37]. Water content in the range of 20–80% was investigated. Figure 7 shows the effect of water content on the immobilized lipase. DES improved the relative enzymatic activity up to 3–4-fold in the range of 20–80% water content. Nevertheless, introducing the DES/AC complex to the reaction media improved the relative activity up to 2–4-fold in the range of 20–40%. Overall, increasing the water content of DES slightly increased the relative activity after 40%.

Experimental observations confirmed the trend of DES activity after an hour of incubation. DES, without any ratio of water added, can enhance the enzymatic activity to a certain level. However, introducing water at a certain ratio to any enzymatic reaction such as hydrolysis reaction may contribute to catalytic enhancement [31]. Previous work [38] has also confirmed that the DES itself, rather than its fragmented components, influenced the enzyme’s catalytic properties. It is proposed that the DES solubilized in aqueous solution still exists, persisting as an intact cation-anion–HBD complex, and this is mainly due to the strong ionic contacts between the cation and the anion, and hydrogen-bond interactions between the anion and the hydrogen bond donor (HBD). Furthermore, in a molecular simulation study, the addition of water to alanine resulted in a significant increase in backbone mobility and a decrease in the compactness of lipase structures, which became more obvious for the open conformation, at 373 K, and high water levels [33]. Overall, their findings suggest that alanine-based DES could be a promising solvent, particularly for the application of lipase at high temperatures. While increasing the water content of alanine slightly increased the backbone flexibility, hydrated DES, especially at low levels (concentration of DES 30 wt.%), still exhibited thermo-stabilizing properties.

#### 2.2.3. Kinetics Parameters

Further experiments were carried out to observe how AC with DES affected the kinetic parameters as contrasted to the buffer medium (control sample). Table 2 summarizes the findings. Measurements of enzyme activity with different concentrations of the substrates were used to evaluate the K_M_ and V_MAX_ values of the reaction of the hydrolysis of pNPP. The non-linear Michaelis–Menten curve was used to determine the parameters with which to express the enzyme kinetics.

The kinetic parameters were determined based on the optimum water content value and the optimum temperature for porcine lipase. Kinetic parameters were computed at different concentrations of the substrate (pNPP), with the parameters being 2.5, 2.0, 1.5, and 1.0 mM. The kinetic parameters, maximum velocity (V_MAX_), and Michaelis–Menten constant (K_M_) were obtained after plotting the kinetic curve with a hyperbolic equation using the software GraphPad Prism. The catalytic measures of the turnover number (k_cat_) and catalytic efficiency (k_cat_/K_M_) were computed, as the amount of enzyme used was known, see Figure 8.

The kinetic parameters V_MAX_ of the porcine enzyme (PPL) and porcine enzyme immobilized on AC in DES as a medium were also estimated at 0.43 and 2.80 mM min^−1^, and the K_M_ was calculated as 0.78 and 4.79 mM, respectively. The K_M_ for the immobilized porcine lipase on AC was 4.5-fold lower than the porcine lipase in phosphate buffer (free enzyme). This indicates that the immobilization of PPL on AC increases the affinity of porcine lipase towards the substrate, which may be due to minor changes caused by electrostatic, hydrogen bonding, or the adsorption of PPL on the AC surface. Moreover, the V_MAX_ value for immobilized porcine lipase in DES medium was higher than that for free porcine lipase, and the activity significantly increased. The results agree with those obtained by Elgharbawy et al. [39] using dendrimers nanomaterials for lipase activation.

A study by Ramani et al. [40] dealt with mesoporous activated carbon (MAC) surface functioning, employing ethylene-amine and glutaraldehyde for strong acidic lipase (AL) immobilization on the MAC. AL has been made by employing lipid waste and fermentation by Pseudomonas gessardii. This study shows that the K_M_ value of the immobilized lipase (0.21 mM) was lower compared to the free lipase before immobilization (0.66 mM) [40]. These results agree with our findings. In addition, two different techniques were employed by Soni et al. to immobilize Burkholderia cepacia lipase treated with surface carbon nanofibers, adsorption and covalent immobilization [41]. They calculated the kinetic parameters and found that K_M_ was 2.07 mM and V_MAX_ was 1.73 mmol min^−1^ g^−1^ for the free lipase. In contrast, the values recorded for immobilized lipase were 0.15 mM and 1.77 mmol min^−1^ g^−1^, respectively [41]. In comparison to the results obtained in our study, the data are analogous since the activated carbon did not alter the V_MAX_. However, the presence of the DES increased the value of activity significantly.

The overall efficiency calculation revealed that the catalytic activity of immobilized PPL with AC is two and four times higher than both pure PPL and PPL immobilized with AC in DES medium. This shows the promising capacity of the activated carbon in the enzymatic performance, although the velocity of the reaction was higher in the DES medium. However, the findings could be slightly different if other types of DESs were to be used in the study. It is to be noted that the study aimed not to investigate the effect of DESs, but rather to find the interaction of lipases with activated carbon and DESs.

### 2.3. Morphology of Activated Carbon

The surface area and pore characterization of AC was measured. Figure 9 shows the N_2_ adsorption–desorption isotherm for AC, and the result showed that the micropore volumes are 0.047 cm^3^/g, and the surface area was 117.447 m^2^/g. This is conducive to provide more active sites for the lipase immobilization. The micropore volume structure was also confirmed by the fact that the produced AC had a micropore volume of 0.047 cm^3^/g, which is suitable for lipase adsorption. Islam et al. [28] found that AC from rattan hydrochar had an average pore width of 3.55, confirming the trend of NaOH activation to form mesoporous carbon. It was hypothesized that increasing the activator-to-hydrochar ratio would increase the activator’s etching depth on the hydrochar surface, converting more micropores to mesopores.

The surface morphology of activated carbon was observed under scanning electron microscopy (SEM). The morphology of the activated carbon surface 1000 magnified is shown in Figure 10.

The surface of activated carbon development and uniformity were both enhanced by NaOH activation. The formation of a well-organized pore structure was accomplished. The shape, size, and structural changes in enzymes and supporters are frequently determined by identifying and characterizing the supporter [6]. The good mechanical properties and the large surface area can hold enough enzymes with minimum diffusion [42].

Hydrochar’s C-O-C and C-C bonds are broken during the activator’s dehydration effect, leading to the formation of AC porosity. As a result of this reduction process, the sodium hydroxide (NaOH) is transformed into sodium metal, sodium carbonate (Na_2_O_3_), and hydrogen gas (H2), as demonstrated by Equation (3)
(1)$$6\mathrm{NaOH}+2C\overset{\Delta }{\to }2\mathrm{Na}+3H_{2}+2{\mathrm{Na}}_{2}{\mathrm{CO}}_{3}$$

The breakdown of Na_2_CO_3_ into Na, CO, and CO_2_ results in additional activation when N_2_ gas is present and the temperature is high. As a result, more Na molecules become diffuse in the pores, widening the holes and forming new pores [31].

## 3. Materials and Methods

### 3.1. Chemical and Biochemical Materials

Palm racemes were utilized as a starting raw material to produce the activated carbon. Palm raceme samples were collected from a variety found in the west of the Baghdad area of Iraq. To remove dust and impurities, the raceme was washed several times in purified water before drying at 100 °C and 24 h, being crushed with a knife’s Mill, and being sieved. For the preparation, a portion with an average particle size of 5 mm was chosen. Palm raceme was chemically activated with sodium hydroxide and hydrochloric acid (Poch SA Company, Poland). In addition to sodium deoxycholate (purity 97%), isopropanol, and Arabic gum, Sigma-Aldrich provided the standards and substrate for the lipase assay: *p*-nitrophenol (*p*NP) and *p*-nitrophenyl palmitate (*p*NPP). Lipases from the porcine pancreas (PPL) (100 to 500 units, mg^−1^ protein) and Amano lipase PS originated from *Burkholderia cepacia* (AML) (30 units, mg^−1^ protein) and were acquired from Sigma-Aldrich. HmbG Chemicals and MERCK (Hamburg, Germany) provided additional analytical-grade chemicals.

### 3.2. Preparation of Activated Carbon (AC)

Different impregnation ratios (the ratio of activator to dried palm raceme weight) were used to combine 10 g of dried palm raceme with 100 mL of NaOH solution. The concentration of NaOH depends on the impregnation ratio (IR), where 100 mL of distilled water was added to dry NaOH. The impregnation mass ratios of NaOH to biomass (palm raceme) used in this study were 0.5, 1.0, 1.5, 2.0, and 2.5. The mixtures were then gently heated for half an hour at 80 °C during mixing and then kept for 24 h at room temperature. The treated samples were then dried in an oven (Model IH-100, England) at 100 °C until totally dry and kept in a desiccator. A cylindrical stainless-steel reactor was used to carbonize dried impregnated samples. In an electric furnace, the reactor was heated at a continuous rate of 10 °C min^−1^ and kept (during 0.5–2.5 h) at carbonization temperatures (400–800 °C). The samples were removed from the furnace and allowed to cool at the end of the activation process. The samples were then immersed in a 0.01 M HCl solution for 10 min, resulting in 10 mL·g^−1^ as a liquid to solid ratio. The mixtures were placed overnight at room temperature and washed several times with distilled water before the filtrate pH reached 6.5–7. The samples were dried at 100 °C for 24 h afterward. The impacts of impregnation ratio (0.5–2.5, NaOH/biomass, *w*/*w*), activation temperature (400–800 °C), and activation time (0.5–2.5 h) of activated carbon on lipase activity were investigated to optimize the preparation conditions of carbon activated with NaOH. Table 3 lists the experimental runs and conditions for preparing AC using NaOH activation. Finally, the samples were placed in tightly-sealed bottles for further analysis.

### 3.3. Preparation of DESs

The DES was prepared from alanine (89.1 g) and NaOH (40 g), which were mixed for two hours at 350 rpm at 85 °C until the DESs appeared to be in the homogeneous phase. For future use, the prepared DESs were stored in a regulated moisture state.

### 3.4. Immobilization of Lipases

AML, and PPL were immobilized on 0.1 g of activated carbon that was treated at different conditions, namely A1 to A13. Lipases solution (0.5 mg/mL) was incubated with 0.1 g of AC sample for 2 h at 350 rpm in a thermomixer (BIOBASE), and the activity was then assessed using the lipase assay as illustrated in Section 3.5.

### 3.5. Assay for Lipase Activity

A colorimetric method was used to assess the activity of the lipase enzyme in aqueous solutions. To measure lipase activity, the *p*NP which was liberated due to the hydrolysis of *p*NPP was chosen as a standard approach for assessment. Substrate solution A contained 7 mg of *p*NPP dissolved in 4 mL isopropanol. Solution B contained 0.07 g of Arabic gum and 140 mg of sodium deoxycholate in 65 mL of sodium phosphate buffer (pH 8.00) with 0.35 mL of Triton-X-100 loaded to eliminate turbidity. The final solution was always prepared instantly by mixing solutions A+B. The lipase solution was prepared by dissolving 5 mg of each enzyme in 10 mL of buffer solution (pH 8.00). The reaction was begun by the addition 0.3 mL of lipase, or lipase/DES to 0.1 g of AC, followed by the addition of 0.7 mL of the newly formulated solution. The same amount of the solid lipase was dissolved when preparing the lipase solution in DES and for the purpose of immobilization.

Using a water bath heated to 40 °C, all tubes were incubated for 15 min before the reaction was terminated with a 0.3 mL solution of acetone/ethanol (1:1). A Multiskan^TM^ GO microplate spectrophotometer was used to measure the absorbance of *p*-nitrophenol (*p*NP) at 410 nm. Each assay was performed three times. At 410 nm, the absorbance of *p*NP was determined to calculate the molar extension coefficient (ϵ) from the *p*NP calibration curve (Equation (1)). A unit of enzyme activity is described as the enzyme quantity that releases *p*NP (µmol/min/min) = unit/mL. The DES absorbance without an enzyme was deducted from the actual data for value adjustment.
(2)$$Lipase\;Activity\;\left(\frac{U}{mL}\right)=\frac{\mu \mathrm{mol}\;\mathrm{of}\;p\mathrm{NP}\;\mathrm{realease}\times \;\mathrm{Total}\;\mathrm{reaction}\;\mathrm{volume}}{\mathrm{volume}\;\mathrm{used}\;\mathrm{in}\;\mathrm{spectrophotometer}\;\times \mathrm{enzyme}\;\mathrm{volume}}\times \frac{1}{\mathrm{reaction}\;\mathrm{time}\;\left(\min\right)}$$

Relative activity of lipase was calculated based on Equation (2):(3)$$Relative\;Activity\%=\frac{\mathrm{Residual}\;\mathrm{activity}\;\left(\mathrm{Final}-\mathrm{Initial}\right)U/\mathrm{mL}}{\mathrm{Initial}\;\mathrm{Activity}\;U/\mathrm{mL}}\times 100$$
where Initial is the activity of lipase prior to treatment (the control), Final is the value of the lipase activity after treatment, and Residual activity of the enzyme is the difference between the final value after treatment and before the treatment (control).

### 3.6. Screening of Lipases Activities with Different Activated Carbon

With the thirteen AC samples mentioned in Table 3, two types of lipases, PPL and AML, were screened. In a 2-mL centrifuge tube, the buffer was combined with 1 mL of the free lipase or AC/lipase samples. In the control samples, lipase solution that was prepared in phosphate buffer was used with a 1:1 ratio (1 mL of enzyme at 0.5 mg/mL in 1 mL of buffer), while for the DES/lipase samples, a 1:1 ratio was used with 1 mL enzyme solution against 1 mL DES. All samples were vortexed before proceeding with the following steps. All samples were labeled from A1 to A13 and then accordingly loaded with 0.1 g of AC. The solutions were incubated for 2 h at 350 rpm in a thermomixer (BIOBASE). The supernatant was collected after centrifuging for 2 min (8000 rpm). The supernatant samples were then subjected to lipase activity determination as described in Section 3.5.

### 3.7. Statistical Analyses

Data were expressed as mean ± standard deviation of triplicate analyses. Analysis of variance (ANOVA) was conducted using GraphPad Prism version 9.0.0 to determine the significant difference between the means at a 95% confidence level (*p* < 0.05).

### 3.8. Morphology of Activated Carbon

The surface characteristics of AC were tested by N_2_ adsorption–desorption isotherms at 77 K under varied relative pressure levels. The surface area of the selected AC was determined by the Brunauer Emmett and Teller (BET) surface area analysis using a fully automated gas sorption system (Micromeritics ASAP2020, Plus 2.00) as described by [26]. The BET method was used to calculate the surface area, and the t-plot method was used to calculate the volumes of the micropores and mesopores. Scanning electron microscopy (SEM) was also utilized to identify the surface morphology of AC.

## 4. Conclusions

In this study, AC prepared from palm raceme was successfully utilized for the immobilization of AML and PPL. Alanine-based DES was used for the first time as a co-solvent for the enhancement of enzymatic activity. The results showed that AC can significantly increase the enzymatic activity of PPL, performing better than AML. Different samples (13 in total) of AC were treated under different conditions and defined as A1–A13. In order to select the optimum treatment conditions, these samples were used for enzymatic immobilization and for enzymatic activity enhancement. Under optimum treatment conditions of impregnation mass ratio of 0.5, 150 min, and carbonization temperature of 400 °C, the highest relative enzymatic activity was 162.5% for PPL. Different operating parameters such as incubation temperature and water contents were optimized for the selected AC with/without the aid of DES. Immobilized PPL on AC/DES medium at 60 °C optimum incubation temperature achieved 290.2% of relative enzymatic activity. The relative enzymatic activity was improved up to three-to-seven-fold in the range of 20–40% water content. Furthermore, kinetics data revealed that the catalytic activity of immobilized PPL with AC was two and four times higher, respectively, than both neat PPL and PPL immobilized with AC/DES medium. These findings and conclusions are evidence that the immobilization method using AC and DES is an ideal choice for potential implementation, with specific use for biotechnology applications.

## Figures and Tables

**Figure 1 molecules-27-08734-f001:**
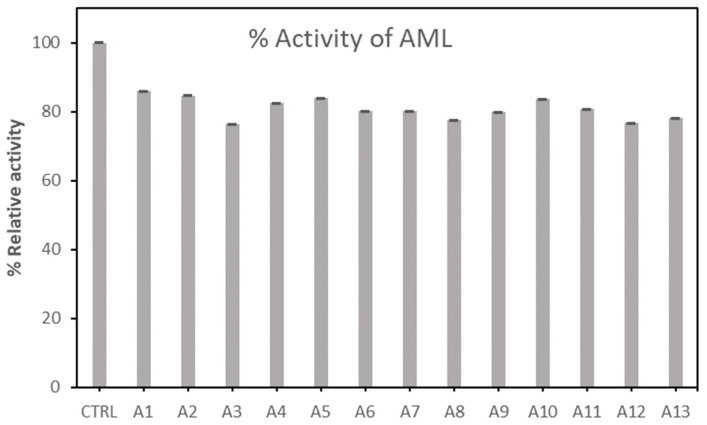
Relative activity of Amano lipase (AML) after immobilization on activated carbon samples from A1 to A13. Data are presented with 95% confidence intervals (error bars).

**Figure 2 molecules-27-08734-f002:**
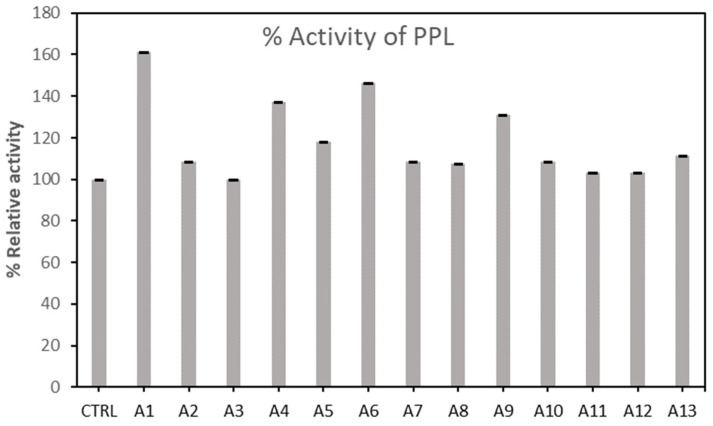
Relative activity of the Porcine enzyme (PPL) after the immobilization on activated carbon samples, from A1 to A13.

**Figure 3 molecules-27-08734-f003:**
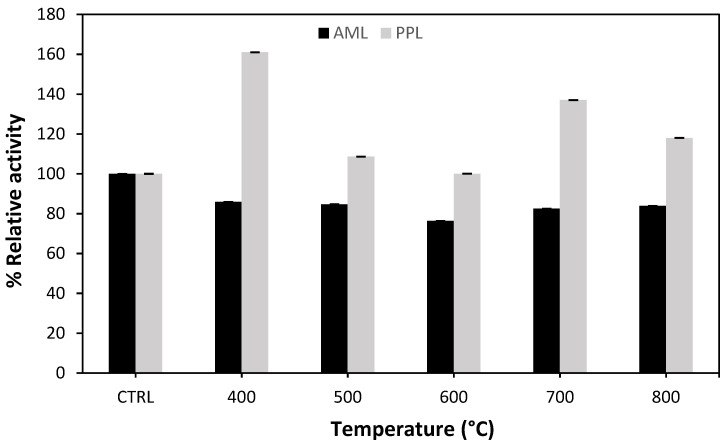
Effect of carbonization temperature of samples A1–A5 on the relative activity of two types of immobilized lipase, Amano lipase (AML) and porcine lipase (PPL), against the free lipases.

**Figure 4 molecules-27-08734-f004:**
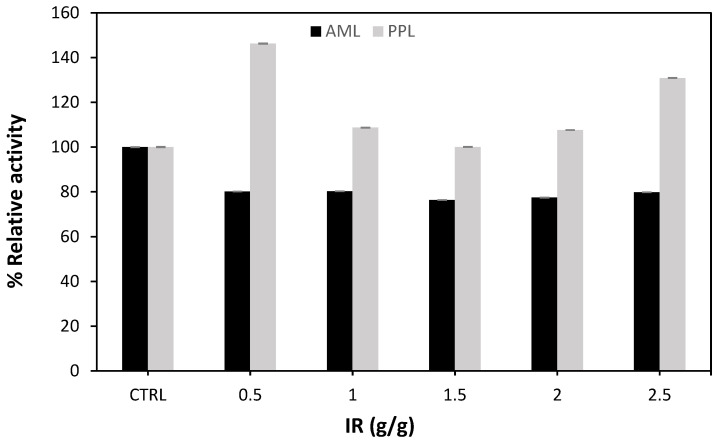
Effect of impregnation ratio (samples A3 and A6–A9) on relative activity for two types of immobilized lipases, Amano lipase (AML) and porcine lipase (PPL), compared to the free enzyme (CTRL).

**Figure 5 molecules-27-08734-f005:**
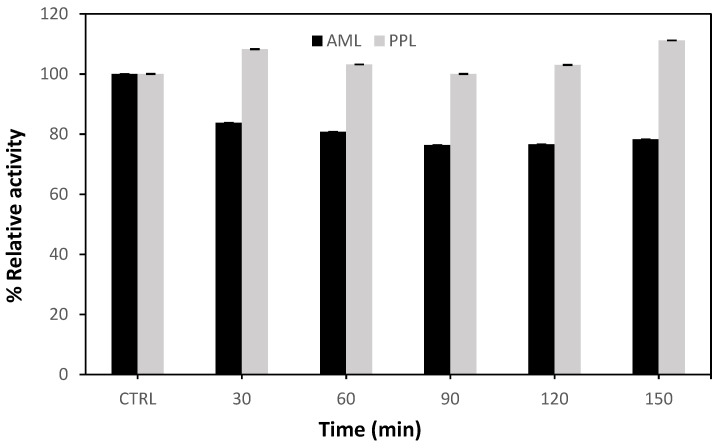
Impact of carbonization time on the relative activity of two types of immobilized lipase, Amano enzyme (AML) and porcine enzyme (PPL), against the free lipases (carbonization Temperature 600 °C and impregnation ratio 1.5).

**Figure 6 molecules-27-08734-f006:**
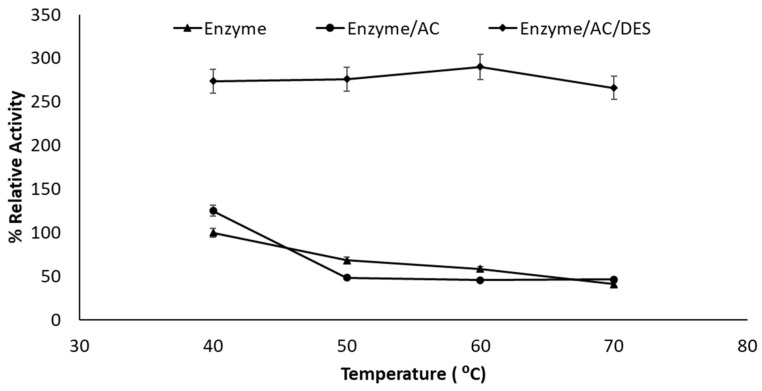
Immobilized and free Lipase activities at various incubation temperatures (2 h incubation time and 1:1 DES molar ratio).

**Figure 7 molecules-27-08734-f007:**
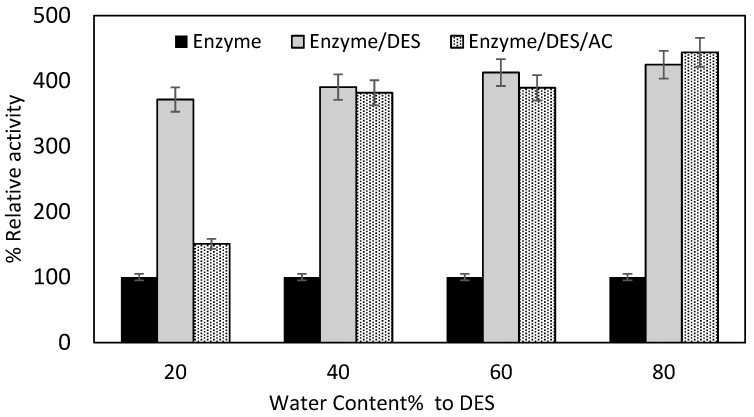
Activity of immobilized and free porcine pancreas lipase (PPL) at different concentrations of aqueous DES (1:1 NaOH: Alanine).

**Figure 8 molecules-27-08734-f008:**
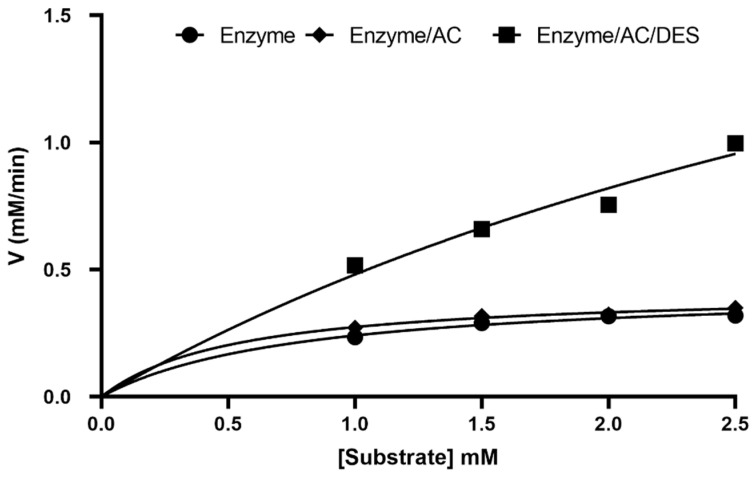
Non-linear kinetic curve (Michaelis–Menten equation) for PPL: free enzyme (control), enzyme with AC, and enzyme with AC/DES, generated by GraphPad Prism version 9.0.0.

**Figure 9 molecules-27-08734-f009:**
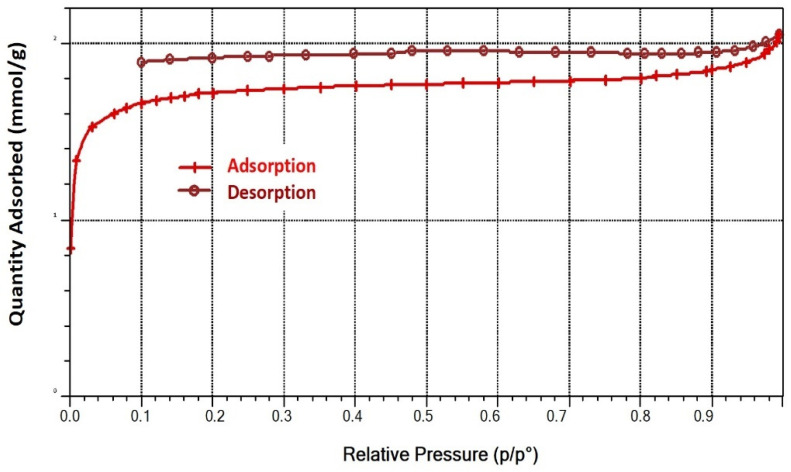
Adsorption–desorption isotherm curves of AC from palm raceme.

**Figure 10 molecules-27-08734-f010:**
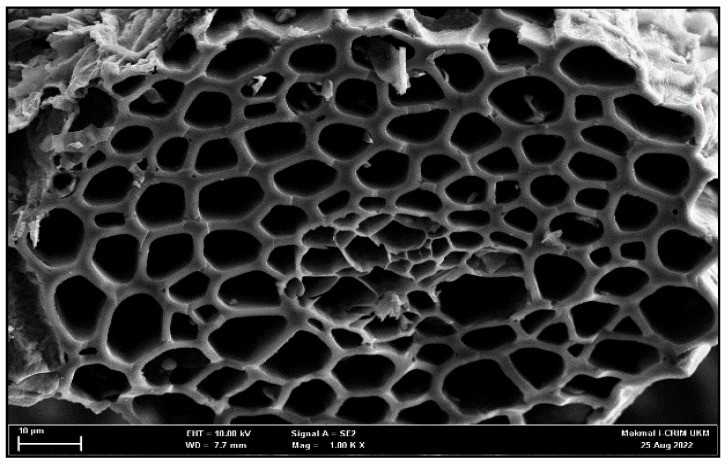
SEM image (at 10 µm scale) of prepared activated carbon at 1000× magnification.

**Table 1 molecules-27-08734-t001:** ANOVA Table for the relative activity of the Amano lipase (AML) and porcine enzyme (PPL) with activated carbon samples, from A1 to A13.

ANOVA Summary	AML	PPL
F	1096	15871
*p* value	<0.0001	<0.0001
Significant diff. among means (*p* < 0.05)	Yes	Yes
R squared	0.9980	0.9999

**Table 2 molecules-27-08734-t002:** Kinetic parameters of immobilized and free porcine lipase (PPL) in various reaction media using *p*NPP at different concentrations.

Medium	K_M_ (mM)	V_MAX_ (mM min^−1^)	K_cat_ (min^−1^)	K_cat_/K_M_ (min mM^−1^)
Enzyme only	0.78 ± 0.03	0.43 ± 0.03	4.01 ± 0.03	5.14 ± 0.01
Enzyme/AC	0.17 ± 0.02	0.42 ± 0.03	3.71 ± 0.02	21.82 ± 0.03
Enzyme/DES/AC	4.79 ± 0.03	2.80 ± 0.02	49.07 ± 0.01	10.24 ± 0.03

**Table 3 molecules-27-08734-t003:** The factorial design of experimental runs and conditions of activation of carbon with NaOH.

No.	Sample	Activation Temperature/°C	Impregnation Ratio (NaOH/biomass)/(g/g)	Activation Time/min
1	A 1	400	1.5	90
2	A 2	500	1.5	90
3	A 3	600	1.5	90
4	A 4	700	1.5	90
5	A 5	800	1.5	90
6	A 6	600	0.5	90
7	A 7	600	1.0	90
8	A 8	600	2.0	30
9	A 9	600	2.5	90
10	A 10	600	1.5	30
11	A 11	600	1.5	60
12	A 12	600	1.5	120
13	A 13	600	1.5	150

## Data Availability

Not applicable.

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
