# Peer review of "Palm Raceme as a Promising Biomass Precursor for Activated Carbon to Promote Lipase Activity with the Aid of Eutectic Solvents"

_molecules, 2022, doi:10.3390/molecules27248734_

Round 1

Reviewer 1 Report (Previous Reviewer 2)

The subject of this manuscript Palm Raceme as a Promising Biomass Precursor for Activated 2 Carbon to Promote Lipase Activity with the Aid of Eutectic Sol- 3 vents is important to readers, and I recommend accepting it after the comments below:

1. Please enhance the first two sentences from the abstract. 2. Modify the introduction, focus on your area, and add the latest studies. 3. Improve the result and discussion. 4: Generally, the English language needs to be improved; so many mistakes were found. 5. In all your tables, please add the expected error. 6. Make sure all references are in the same style following the Journal.

Author Response

Response to reviewer #1 comments

English language and style

() English very difficult to understand/incomprehensible
() Extensive editing of English language and style required
(x) Moderate English changes required
() English language and style are fine/minor spell check required
( ) I don't feel qualified to judge about the English language and style

Response: The English were improved.

Comments and Suggestions for Authors

The subject of this manuscript Palm Raceme as a Promising Biomass Precursor for Activated Carbon to Promote Lipase Activity with the Aid of Eutectic Solvents is important to readers, and I recommend accepting it after the comments below:

  1. Please enhance the first two sentences from the abstract.

Response:

Thank you for the comment. The abstract was enhanced, and the manuscript was revised accordingly.

  1. Modify the introduction, focus on your area, and add the latest studies.

Response:

The introduction of the manuscript was modified and revised.

  1. Improve the result and discussion.

Response:

The results and discussion section were revised according to the reviewers’ comments.

  1. Generally, the English language needs to be improved; so many mistakes were found.

Response:

The English language was improved.

  1. In all your tables, please add the expected error.

Response:

The tables in the manuscript were revised and the expected error was added.

  1. Make sure all references are in the same style following the Journal.

Response:

The references were arranged according to the journal style.

Reviewer 2 Report (New Reviewer)

The submitted manuscript by Khalid M. Abed and coworkers has nicely described the preparation of activated carbon (AC) from palm racemes followed by utilization for the improvement of enzymatic activity with the assistance of DES. The manuscript can be accepted after addressing following minor comments.

1.     SEM image should be in Figure 11 instead of 9; the typographical error needs to be corrected.

2.    The distribution curve in Figure 9 depicted nanometer sized pores whereas from the SEM image at Figure 11, pore sizes are in the micrometer range from the scale. Explain why there are discrepancy.

3.     Authors need to explain more clearly how the pore volume and quantity of AC alter the enzymatic efficiency.

Author Response

Response to reviewer # 2 comments

The submitted manuscript by Khalid M. Abed and coworkers has nicely described the preparation of activated carbon (AC) from palm racemes followed by utilization for the improvement of enzymatic activity with the assistance of DES. The manuscript can be accepted after addressing following minor comments.

  1. SEM image should be in Figure 11 instead of 9; the typographical error needs to be corrected.

Response

The figure of the SEM image was revised.

  1. The distribution curve in Figure 9 depicted nanometer sized pores whereas from the SEM image at Figure 11, pore sizes are in the micrometer range from the scale. Explain why there are discrepancy.

Response

We thank the reviewer for the feedback and constructive comments. Yes, we do agree that there are differences in terms of size between Fig 9 and Fig 11.

After checking we may give the reason mainly due to the analysis of different samples that we tested.  In order to make the study more logical and more concise we did the test again for the same sample of SEM and we found it is in the micrometer range, this part was revised in the manuscript accordingly.

  1. Authors need to explain more clearly how the pore volume and quantity of AC alter the enzymatic efficiency.

Response

The enzyme was immobilized successfully by activated carbon.  The pores and active sites on the activated carbon enhance the immobilization of enzymes in the micro- and mesopores. Further, the lipase molecules are bound on the surface of the supporter after immobilization. The immobilization of enzymes minimizes the diffusion limitation required for efficient bio-catalyzed reactions. Therefore, immobilized enzymes possess several advantages compared to free enzymes such as facile handling, enhance activity, and improved thermal stability.

This manuscript is a resubmission of an earlier submission. The following is a list of the peer review reports and author responses from that submission.

Round 1

Reviewer 1 Report

Additional comments and questions are highlighted by yellow and placed in pop-up windows in *pdf file.

All noticed typos are highlighted by blue.

All References highlighted by green should be checked, corrected and edited.

Detailed сomments and questions:

Line 27. Phrase  AC more interacted with PPL in comparison with AMLshould be explained, especially the words “more interacted, and what does it mean (in adsorption, enzymatic activity, stability, others?

Line 28. Phrase  were 0.5 (NaOH /palm raceme, w/w)should be explained. Is 0.5 the ratio of dry NaOH (in g) to palm raceme (in g)?

Line 35. The type of studied reaction should be indicated in Keywords.

The order of Keywords is recommended to change as follows: activated carbons, lipases immobilization, hydrolysis, deep eutectic solvent, kinetics

Line 70-71. Phrase  aliphatic mixtureshould be explained. What is the mixture of substances: hydrocarbons, alcohols, other?

Line 96. What is the concentration of NaOH in the solutions?

Lines 113, Table 1. There is no systematic description of sample preparation conditions. The samples in the table are presented very randomly and chaotic! This leads to a poor understanding of the work.

Lines 126, Table 2. Why is there no DES based on glycine in the table? Table 2 is recommended to be removed, а description in the text will suffice.

Lines 140–141. It is necessary to describe the hydrolysis conditions in more detail, namely, how many g of lipase, lipasePPL/AC, and lipase AML/AC were used in the reaction mixture, indicate the weight of AC samples, volume of the reaction mixture, etc. Was the same amount (in mg) of lipase, soluble (named by the authors as “free”) and immobilized, used in the reaction?

Lines 145. No results found for DES based on glycine.

Lines 145-146. The value of molar extension coefficient has to be given in the kit instructions.

Lines 147. This is a very ancient definition of enzyme activity, when there were no methods for determining the concentration of a protein in a solution. Now the various methods of protein analysis are developed. Obviously, the dimension of quantity is gram, the dimension of enzyme activity is μmol ·min–1 (the rate of reaction in fact).  Did you determine the concentration of protein in the solution and by what method?

Lines 153. What is the concentration of lipase in solution and what is the value of adsorption of lipases (L) on activated carbon samples? The above designations are L/AC and L/AC/DES. Designations in the text must be the same! Section 2.6 should be rewritten. It is absolutely not clear where the enzymatic activity is measured, in the supernatant, or in the reaction mixture in the presence of L/AC?

Line 166. The aactivities should be measured and the results should be compared with the same amount of soluble and immobilized enzyme.

Line 169, 175, Fig.2. Explain what a heat map is, how and with what program it is built.

It is necessary to decipher the designation CTRL. Why is such a heat map not provided for the PPL?

Fig. 2–7. It is very necessary to give definitions to such notions as total activity of immobilized lipase (U per 1 g of L/AC) and specific activity (U per 1 mg of lipase adsorbed on AC). Also, it is necessary to give the absolute values of these activities (similarly as activity of AML 30 U/mg, line 91). Then, it is very necessary to indicate how the relative activities (in %) are calculated. It would be more correct to calculate the ratios of the specific activity of soluble and immobilized enzymes at exactly identical reaction conditions.

Lines 205-205, 217, 218. It is recommended to measure the parameter such as Brunner Emmet and Teller (BET) surface area and total pore volume of the studied sample and introduce these parameters in Table 1.

Table 3 and Table 4 may be removed. The data presented in these Tables may be described in the text.

Lines 217-218. Prove these statements by determining numerical values of the specific surface area and pore volume.

Lines 236-245. Since the activities of the two immobilized lipases (PPL and AML) are measured under exactly identical condition, these common considerations about hyperactivation do not explain the differences observed. But Reviewer is not sure about the experimental conditions which have to be exactly identical, primarily in terms of the amount of the enzyme (soluble and immobilized) involved in the reaction.

Fig. 6. Edit this figure. Designations in the text and in figures must be the same throughout the paper, L/AC and L/AC/DES as above.

Changes in values on one each curve look unrepresentative (for example, for L/AC). What is the experimental error? Changes in activity above 50oC are possible within the error.

Line 311, 314,315. It is unclear what does the term “reline” mean.

Line 327, 333. Lineweaver-Burk plot is common sufficiently old approach to determine kinetics parameters. The current trend is to programmatically approximate the full kinetic curve with a hyperbolic equation.

Show graphically 4-point linear fit to prove that the curve is linear. It is very important!

Line 338, 339. The number of significant digits in the results should be adequate to the experimental error. Seven (7) significant digits is clearly redundant.

Line 378. According to Table 5, the catalytic efficiencies are the same, equal to 5.6 min mM-1.

Conclusions need to be rewritten.

Author Response

We thank the reviewers for the feedback and constructive comments. Attached below we have responded to the comments accordingly. Please see the attachment.

Reviewer 2 Report

The manuscript is well written and has good knowledge, which is very important, I recommend accepting it after minor revision.

    1. The keyword is not suitable. Please change it to conflict with the content in general and to be more visible for researchers. 2. In the introduction, please focus more on your work. 3. Please add more explanation in the result and discussion along with the newly related reports. 4. Make sure all references are in the same style.

Author Response

(The authors gave the same response as above.)

Round 2

Reviewer 1 Report

Line 232-233.  Let’s consider the Equation 1.

The total reaction volume is equal to 1.0 mL, the lipase volume is 0.3 mL of solution at 0.5 mg of protein per 1 mL of buffer (the amount of protein is 0.15 mg).  The ratio of volumes is 3.3 (dimensionless).

Lambert-Bouguer's law is D=C·l·?, where D – absorbance (optical) density, C – concentration, mol/L, l - cuvette length, sm. It is unclear what dimension has ?, equal to ca. 4.5.

1 U = μmol/min. Activity in Equation 1 is expressed in U/mL. Absolutly unclear!

Line 271-272.  Indicate what the residual, final and initial activities mean. Initial and final activities were measured before and after adsorption of lipases on activated carbons, or not?

Figures 1 and 2 are identical, but the designation of the samples in Table 1 has changed. For example, A1 (previous) – 600oC, 1.5, 150 min, A1 (new) – 400oC, 1.5, 90 min, or
A13 (previous) – 500oC, 1.5, 90 min, A13 (new) – 600oC, 1.5, 150 min.

Line 423-424.  Checking Required! Are AC samples macroporous with an average pore diameter of 1.2 µm and a surface area of 417 m2/g? Please submit a pore size distribution diagram! If AС samples are macroporous, then reasoning about the accessibility of the surface areas during lipase adsorption is inappropriate.

Line 443-452.  Unfortunately, reasoning about the structure of the lipase lid is not related to the topic of the research. These lines may be deleted without damage of the theme of the research

Comments, questions, noticed typos are highlighted by yellow and placed in pop-up windows in *pdf file.
